# Effectiveness of Foodbank Western Australia’s *Food Sensations*^®^ for Adults Food Literacy Program in Regional Australia

**DOI:** 10.3390/ijerph18178920

**Published:** 2021-08-25

**Authors:** Catherine Dumont, Lucy M. Butcher, Frances Foulkes-Taylor, Anna Bird, Andrea Begley

**Affiliations:** 1Foodbank of Western Australia, Webberton Road, Geraldton, WA 6530, Australia; catherine.dumont@foodbankwa.org.au; 2Foodbank of Western Australia, Abbott Road, Perth Airport, Perth, WA 6105, Australia; frances.foulkes-taylor@foodbankwa.org.au; 3School of Public Health, Curtin University, Kent Street Bentley, Perth, WA 6102, Australia; anna.bird@curtin.edu.au (A.B.); A.Begley@curtin.edu.au (A.B.); 4Telethon Kids Institute, Nedlands, WA 60009, Australia

**Keywords:** food literacy, nutrition education, regional, remote

## Abstract

Background: *Food Sensations* for Adults, funded by the Western Australian Department of Health, is a four-week nutrition education program focused on food literacy, with demonstrated success amongst Western Australians. In the last two years, 25% of programs have been in regional and remote areas and therefore the aim of this research is to explore the impact of the program in regional areas. Methods: Participants answered validated pre- and post-questionnaires to assess change in food literacy behaviours (2016–2018). Results: Regional participants (n = 451) were more likely to live in low income areas, have lower education levels, and identify as Aboriginal, than metropolitan participants (n = 1398). Regional participants had statistically significantly higher food literacy behaviours in the plan and manage and preparation domains, and lower selection behaviours at baseline than metropolitan participants. Post program, regional participants showed matched improvements with metropolitan participants in the plan and manage, and preparation domains. Food selection behaviour results increased in both groups but were significantly higher in regional participants. Conclusions: The program demonstrates effective behaviour change in all participants; however, the increased disadvantage experienced by people residing outside of major cities highlights the need for additional government support in addressing regional specific barriers, such as higher food costs, to ensure participants gain maximum benefit from future food literacy programs.

## 1. Introduction

Poor diet, particularly inadequate fruit and vegetable intake and overconsumption of discretionary foods, is a leading modifiable risk factor contributing to growing chronic disease rates in Australia [1]. The dietary intakes of regional Australians differ to those in metropolitan areas [2]. The 2017–2018 National Health Survey found that adults living in regional areas were more likely to meet vegetable recommendations, but not fruit, when compared to those living in major cities (9.5% with 6.9% respectively) [2], however, they were also more likely to consume sugar-sweetened drinks daily (14% compared to 8.3%) [2]. The dietary differences are a contributing factor to regional Australians being more likely to be overweight or obese compared to people in major cities (72.4% with 65.0%, respectively) [2]. 

The factors affecting poorer dietary intakes and health outcomes in regional Australians are varied and complex. Those living outside metropolitan areas are more likely to be socioeconomically disadvantaged, with lower incomes, education levels, and employment opportunities [3]. Reduced food security is another factor contributing to the disadvantage experienced by regional Australians [4,5,6]. Food security (i.e., the reliable access to sufficient, affordable, and nutritious food) is impacted by socioeconomic, geographical, and environmental determinants. In Australia, food prices increase and household incomes decrease proportionally with distance from major cities [5], resulting in reduced food affordability for those living in regional areas [5,7]. Physical access to food is dependent on households’ distance to, often geographically scarce, food outlets, as well as reliability of transportation, and availability of stock in food stores [5,6]. The availability, cost, quality, and shelf life of perishable food items in regional food stores is further impacted by freight, which is often over vast distances, and in harsh and unpredictable weather conditions [5,7,8]. 

Additional to availability and access, another key dimension of food security is utilisation, which encompasses food literacy, which is the knowledge, skills, and behaviours required to plan and prepare healthy, affordable food [9]. A food literate person is thought to have greater resilience against food insecurity through the ability to employ multiple coping strategies to maximise limited resources and reduce impact on dietary intake [9]. For this reason, food literacy programs are frequently targeted at disadvantaged populations, such as people residing in regional areas, in the view of building self-efficacy and skills to improve dietary intakes [10]. 

There is limited Australian evaluation of nutrition education programs focusing on food literacy, especially in regional Australian communities [11]. Only two programs, primarily focused on cooking skills, have shown positive dietary and social behaviours in the regional Australian context. These programs demonstrated increased participant cooking skills, self-efficacy and confidence, and self-reported increases in vegetable intake [12,13]. From an international perspective, there are only a few studies [14,15] that aimed to increase nutrition and cooking skills in regional adult populations, but to date no studies have compared the impact of a food literacy program on both metropolitan and regional participants. 

Foodbank of Western Australia (WA) has invested in food literacy programs since the mid-1990s in an effort to improve food security and build dietary reliance of disadvantaged populations [16,17]. Foodbank WA’s *Food Sensations*^®^ for Adults (FSA) is promoted as an adult nutrition education and cooking program targeting individuals from low- to middle-income households, who would like to increase their food literacy. The FSA program is delivered by trained health professionals, and has a proven beneficial impact on food literacy behaviours and dietary intake both immediately after completion [18] and at three months follow-up [19]. First implemented in 2011, FSA underwent extensive redevelopment in 2015 to align with an Australian Food Literacy Model [9], the latest government dietary guidelines [20], and the Western Australian Department of Health’s Best Practice Criteria for Food Literacy Programs [21]. Funded by the Western Australian Department of Health, FSA is promoted as a free nutrition education and cooking program. Participants are recruited through existing community groups, or are able to self-enrol in public programs. 

FSA is a four-session, experiential nutrition education program, equating to ten hours of contact time for each program participant (Figure 1). FSA’s curriculum comprises eight lesson plans that are divided into four core modules (taught over the first three sessions), and several optional modules, of which participants may select one (60 min) or two (30 min) to be taught in the fourth and final week. All core module content has been mapped to the four domains (Planning and Management, Selection, Preparation and Cooking, and Eating) and 11 components of food literacy outlined in the empirically tested Australian Food Literacy Model [9]. Optional modules are offered in session four, to reinforce the food literacy key messages and to enable the contextualisation of content to meet the needs of various different subgroups of participants [22]. 

The FSA program is contracted to service the entire state of WA, an area of over 2.5 million square kilometres or 11 times the size of the United Kingdom [23]. Multiple strategies are employed to increase access to the FSA program. Regional face-to-face program delivery is primarily undertaken by two Foodbank WA public health nutritionists; one based in Peel, 75 km south of Perth, who services the South West region, and one based in Geraldton, 425 km north of Perth, who services the Mid West region. Public health nutritionists located at the Perth Foodbank branch also travel to regional areas or use innovative video conference (VC) technology to extend FSA’s reach to other regional locations. Video conference programs are hosted by local Community Resource Centres (CRCs). In partnership with Foodbank WA, these not-for-profit centres are independently operated by the local community and provide the conference and kitchen facilities used to deliver the program to residents in the surrounding areas. All metropolitan programs were delivered in a face-to-face format. Emerging evidence suggests that video conference or online program delivery is as effective as face-to-face delivery [24,25]. For this reason, the results from both delivery formats are considered together.

The FSA contract requires the delivery of 100 programs per annum, with 25% of delivery in regional areas. Of all regional annual FSA delivery, approximately 30% of the programs are delivered by videoconference, and 70% are delivered face-to-face. Additionally, WA Country Health Service (WACHS) health professionals are trained through *Food Sensations* training, delivered by FSA team members. Once trained, health professionals are able to deliver FSA (face-to-face) in their service areas, on behalf of Foodbank WA and provide delivery data after program completion.

The aim of this research was to determine if there are differences in the effectiveness of FSA in regional and metropolitan participants. The objectives were to (1) assess if there were differences between regional and metropolitan participants at program enrolment in food literacy behaviours, selected dietary behaviours, and demographic characteristics, and (2) establish if regional participants made similar improvements in food literacy behaviours, serves of fruits and vegetables, and frequency of fast food and sugar-sweetened drink consumption, by completion of the program. For the purpose of this research, regional is defined as any area outside of a major city in Western Australia. This includes inner and outer regional (rural) areas, and remote and very remote areas. The term metropolitan encompasses all major cities as defined by the Australian Bureau of Statistics [26].

## 2. Materials and Methods

### 2.1. Study Design

Participants attending FSA programs run between May 2016 and June 2018 were invited to complete validated [27] pre- and post-program paper-based questionnaires prior to starting the first session, and on completion of the last session (n = 1849). Post-program questionnaires were administered online or over the phone if participants did not complete the paper-based version at the final session. Of the participants who provided some evaluation data, 24.4% (n = 451) were from regional areas covering 24 different towns in 59 programs. Most regional participants were recruited from existing community groups (42.8%), joined a video conference program through their local Community Resource Centre (30.4%), or attended an open-to-public program (26.8%). Not all programs and/or participants were evaluated for various reasons, including limitations relating to mental health, disability, low English language proficiency, or lack of participant consent. There was no reimbursement for completing questionnaires.

### 2.2. Questionnaire Design

The items for the pre- and post-program questionnaires, developed to address the funder’s required service-level outcomes, included a 14-item food literacy behaviour checklist, and four short questions on dietary behaviours to measure change. The pre-program questionnaire included additional items including four food literacy-related practices, a question on reasons for enrolment, and eight sociodemographic variables. The development and validation process for the food literacy behaviours checklist has been published elsewhere [27]. The checklist development process included multiple considerations, such as respondent burden, cognitive load, and reading levels of potential participants. Three food literacy-related practice questions included in the pre-program questionnaire were selected from the Western Australia Health Department’s Nutrition Monitoring Surveillance Survey (NMSS) [28]. These questions covered level of household responsibility for choosing and preparing meals and shopping (similar to those used in the US National Health and Nutrition Examination Survey [29]), and self-rated cooking skills drawn from unpublished qualitative research to inform the *Go for 2&5*^®^ fruit and vegetable social marketing campaign [30]. An additional food literacy-related practice question was included on ‘attitude to cost of healthy foods’ to measure one objective required by the funder. Four short dietary questions were adapted from the same survey series, including two questions on average consumption of serves of fruits and vegetables, and two questions on the frequency of consumption of fast food meals and sugar-sweetened drinks. Demographic characteristics collected from participants included sex, age, highest education level, household composition, postcode, birth in Australia, and identifying as Aboriginal and/or Torres Strait Islander. Income as a secondary demographic characteristic was extrapolated from self-reported postcode and converted to the Australian Bureau of Statistic’s Socio-Economic Indexes for Areas (SEIFA) decile ranking of the Index of Relative Socio-Economic Disadvantage [31]. Deciles 1 to 4 are considered low-income, 5 to 7 middle-income, and 8 to 10 high-income. 

### 2.3. Analysis

Previous exploratory factor analysis of questions relating to food literacy behaviour has successfully identified three food literacy behaviour domains: *Plan and Manage*, *Selection*, and *Preparation*, using 11 questions [27]. *Plan and Manage* included ‘Plan meals ahead of time’, ‘Make a list before you go shopping’, ‘Plan meals to include all food groups’, ‘Think about healthy choices when deciding what to eat’, and ‘Feel confident about managing money to buy healthy food’. *Selection* included ‘Use a Nutrition Information Panel to make food choices’ and ‘Use other parts of the food label to make food choices’. *Preparation* questions were ‘Cook meals at home using healthy ingredients’, ‘Feel confident about cooking a variety of healthy meals’, ‘Try a new recipe’, and ‘Change recipes to make them healthier’. Factor scores for each of the three food literacy behaviour domains were calculated for each participant both pre- and post-program. Possible responses to each question, *Never*, *Sometimes*, *Most of the time* and *Always*, were scored one to four, respectively. This response score was multiplied by the factor loading for each question, and the factor score was calculated by summing the values of each included question [32]. 

Data were analysed using SPSS (IBM) version 25 to examine the effectiveness of the program in improving food literacy behaviour scores, and t-tests were carried out using pre- to post-program scores. The majority of responses to food literacy-related variable questions were recorded as categorical values. Self-reported fruit and vegetable intake in serves were coded as continuous variables. To investigate change in self-reported dietary intake, t-tests were used to compare intake pre- and post-program in both fruit and vegetable serves. To explore change in reported frequency of fast food meals and sugar-sweetened drink consumption pre to post-program, McNemars tests were employed.

### 2.4. Ethics Approval

Ethics approval was obtained from the Human Research Ethics Committee at Curtin University (RDHS-52-16). Participants were provided with a verbal explanation of the purpose of the research at the start of their first session and a written research information sheet. Written consent was obtained prior to questionnaire administration. 

## 3. Results

### 3.1. Response Rate and Demographic Characteristics

Questionnaire data were collected from 1849 participants: 1618 pre-program (87.5%) and 1323 post-program (71.6%). The missing data in the questionnaires were random and no questions were commonly missed. Missing data was equal across the two participant groups. Regional participants (n = 451) statistically varied by sex, household composition, education level, and SEIFA index, as well as being born in Australia and identifying as Aboriginal and/or Torres Strait Islander (Table 1). Regional participants were more likely to be female, born in Australia, to have completed some or finished secondary school as their highest education qualification, come from a low SEIFA postcode area, and identify as Aboriginal and/or Torres Strait Islander. Age and employment status did not differ between regional and metropolitan participants. 

### 3.2. Baseline Comparison

Two-tailed t-tests comparing baseline food literacy domains identified that regional participants had statistically significant higher *Plan and Manage* and *Preparation* behaviours and lower *Selection* behaviours at the start of the program (Table 2). The proportion of difference between regional and metropolitan participants was 3.17% for *Plan and Manage*, 5.13% for *Preparation*, and −5.76% for *Selection*. Regional participants also self-reported a statistically significantly higher intake of serves of vegetables at baseline, with a 13% higher intake than metropolitan participants (*p* < 0.0001). There was no statistically significant difference in intake of serves of fruit between regional and metropolitan participants. 

Self-reported fast food meal intake was significantly lower in regional participants at baseline (*p* < 0.05) (Table 3). Fewer regional participants (4.6% compared to 5.6% of metropolitan participants) reported Three or more times a week fast food meal consumption, and Once or twice a week (22.3% regional, 27.8% metropolitan). Regional and metropolitan participants had similar proportions for those reporting Less than once per week (39.7% regional, 39.8% metropolitan). Thirty-three per cent of regional participants reported Never compared to 26.8% of metropolitan participants.

Regional participants differed in their self-reported sugar-sweetened drink intake, as this was significantly higher in regional participants at baseline (*p* < 0.05) (Table 3). Seven and a half per cent of regional participants reported an intake of Five or more times a week, compared to 5.7% of metropolitan participants. 

Regional participants were statistically significantly more likely to have all the responsibility for shopping, have higher self-rated cooking skills, and agree or strongly agree that healthy foods cost more than unhealthy foods at baseline (52.1% compared to 44.1% metropolitan) (Table 4). There was no difference in reporting of running out of money for food in the past month at the start of the program between metropolitan and regional participants, but approximately 40% of participants did indicate a positive response.

### 3.3. Program Impact

A statistically significant increase in post-program scores for all three food literacy domains was identified for metropolitan participants (*p* < 0.0001) (Table 5). The proportion of the score increase in post-program scores compared to pre-program scores was 26% for *Selection*, 14.2% for *Preparation*, and 11.2% for *Plan and Manage*. There was also a statistically significant increase (*p* < 0.0001) in self-reported fruit and vegetable serve intake, increasing by 17% and 29.6%, respectively. This equated to an average increase of 1/4 serve of fruit and 2/3 serve of vegetables.

Similarly, regional participant scores significantly increased for all three factor scores (*p* < 0.0001) (Table 6), with 22.5% for *Selection*, 5.7% for *Preparation* and 5.4% for *Plan and Manage*. Both metropolitan and regional participants report similar post-program means for all three food literacy domains. Regional participants self-reported a statistically significant increase (*p* < 0.05) in fruit and vegetable serve intake, increasing by 9.6% and 10.3%, respectively. This equated to an average increase of 1/6 serve of fruit and 1/4 serve of vegetables.

At the end of the program, regional participants reported significantly improved *Selection* behaviours compared to metropolitan participants (*p* < 0.01) (Table 7). There were no significant differences between behaviours in the domains *Plan and Manage* and *Preparation*. Similarly, there were no significant differences between serves of fruit and vegetable intake at the end of program. 

Self-reported fast food meal intake was significantly different pre- and post-program for metropolitan participants (*p* < 0.0001). Of those reporting Three or more times a week fast food meal consumption pre-program, only 28.6% reported this post-program; 71.4% reported a lower frequency. There was no significant difference pre- and post-program for regional participants (n = 259, *p* = 0.12170). 

Similarly to fast food meal intake, there was no significant difference in pre- and post-program consumption of sugar-sweetened drinks in regional participants, (n = 258, *p* = 0.0612), but there was for metropolitan participants. Of the metropolitan participants who initially reported consuming sugar-sweetened drinks Three or more times a week pre-program, 47.3% reported a decreased intake and 52.7% reported the same intake post-program. 

## 4. Discussion

This is the first study to compare the impact of a food literacy program between regional and metropolitan participants, and contributes to the limited body of evidence about regional food literacy behaviours and the effectiveness of a state-wide program in the Australian context. A strength of the study is that results are derived from a validated food literacy questionnaire [23]. This responds to a known limitation of not using validated surveys in the evaluation of these programs (14). Results from this study highlight the socioeconomic barriers faced by people living in regional areas, which is consistent with other Australian research [5,13,33,34]. While the FSA program demonstrates effective behaviour change in regional participants, the baseline demographic characteristics reveal a higher level of disadvantage for this population group compared to their metropolitan counterparts. Regional participants were more likely to live in less resourced areas and have lower education levels than metropolitan participants. This may account for the finding that, pre-program, regional participants scored lower in the *Selection* domain, which includes tasks that require a higher level of literacy such as label reading. This study also found that after the program, *Selection* was the food literacy domain that improved the most, yet at the same time regional participants did not change their attitude that healthy foods cost more than unhealthy foods. This may reflect the financial barriers faced by people living in regional areas, such as lower household incomes and inflated food prices, and the inability of food literacy programs alone to overcome these barriers [1,5]. 

In contrast at baseline, participants living regionally had greater scores in the *Plan and Manage* and *Preparation* domains when compared to metropolitan participants. The researchers hypothesise that participants residing in regional areas may more frequently demonstrate and use skills within the *Plan and Manage* and *Preparation* domains out of necessity. Those residing in regional areas may need to travel long distances to supermarkets, and therefore buy large quantities of food to reduce the necessity for more frequent shopping trips [5]. The limited options in food choices and food stores in close vicinity to their homes makes the purchase as you go model of food shopping more difficult for regional participants [35]. This is supported by evidence from Canada that those living in regional and rural areas were highly skilled in feeding households on limited economic resources [36]. Responsibility for food preparation and shopping in regional areas, in this study, seems to follow traditional gender roles more closely than in metropolitan areas or the state average [28] and may reflect more conservative attitudes about food choices [37,38]. Female participants in regional areas made up a greater proportion of participants, and may therefore have contributed to the higher reporting of sole responsibility for food shopping and food preparation, which could explain the higher *Preparation* score at baseline. Finally, at baseline, regional participants had significantly higher consumption of vegetables and a lower intake of fast food when compared to metropolitan counterparts. It is possible these differing intake behaviours may be related to reduced availability to take away foods in regional areas, or the cost of eating out was prohibitive, which in turn means that regional participants have to rely more on food preparation skills and home-cooked meals with greater vegetable content. However, the sugar-sweetened drink intake of regional participants was significantly higher than their metropolitan counterparts, potentially indicative of the low cost and ready access to these food items regardless of geographical location [39]. 

Similar improvements in fruit and vegetable intake were reported in both regional and metropolitan areas. A comparison of the results of FSA with other similar programs is challenging due to the limited number of published regional food literacy programs and differing evaluation methods. However, the increase in vegetable consumption was consistent with the findings of a longer six-week cooking-focused program in a large regional town in Queensland [13]. Other research from regional Australia and the United Kingdom found positive dietary attitudes, social and community connections, and high food literacy were associated with greater fruit and vegetable intake [34,40,41]. Both fast food and sugar-sweetened drink consumption did not significantly change in FSA regional participants post program, while sugar-sweetened drink intake significantly decreased in metropolitan participants. 

The results of the study reveal FSA is successful in changing food literacy and dietary behaviours across multiple different geographical settings. FSA’s effectiveness may be attributed to several key strengths. Firstly, the program’s capacity is tailored to meet the needs of a wide range of groups, including people living in regional areas. It is likely that there is value in the social interactions and support in regional programs where there might be limited opportunities to participate in a range of programs [15]. Examples of how FSA’s curriculum is contextualised within regional settings include changing the food prices in activities to reflect local pricing, selecting recipes where all ingredients are readily available, and/or adapting recipes to use more frozen or canned produce. These minor changes are an important means of recognising regional barriers to food access and availability, and guide participants to implement coping strategies to address these challenges [42]. Continued efforts are needed to create healthful food environments, such as addressing issues with the access, availability, quality, and prices of healthy foods, in regional areas to enable individuals to practice skills gained from attending food literacy programs and support long-term positive behaviour change [43]. 

While the FSA curriculum has flexibility built into the lesson content and structure, several processes ensure the program also has a high degree of fidelity. Foodbank WA staff, who are university-trained nutrition professionals, provide regular, comprehensive FSA training to statewide partners, to ensure consistency in the delivery and evaluation of the program, which is in line with national dietary and best practice guidelines. Collectively, these factors improve the generalisability of the program to all of Australia, and possibly to other countries that have similar levels of remoteness. Secondly, the program strives to ensure equity of access to food literacy education irrespective of geographical location. The broad reach of FSA is achieved through regional partnering with the WA Country Health Service and other organisations to increase delivery of the program, employment of two regional Foodbank WA staff, regional travel by the Perth FSA team, and using innovative videoconferencing technology to deliver to regional community centres. 

### Limitations

FSA program findings need to be considered within the context of several limitations. Firstly, all data are self-reported and therefore subject to social desirability bias. Secondly, self-reported food literacy skills relate to a participants’ confidence rather than reflecting their actual cooking abilities and behaviours. Thirdly, while the study did include some indicators of disadvantage, the reported incidence of running out of money for food was similar in metropolitan and regional areas. Therefore, the research may not have captured the full range of disadvantage in regional areas in Western Australia. Lastly, data are cross sectional in nature with no comparison group in place. 

## 5. Conclusions

The findings of this study demonstrate that FSA effectively produces improvements in food literacy behaviours within a four-week period in both regional and metropolitan participants. These findings also highlight the disadvantage faced by regional Australians, such as the perception of the higher cost of healthy foods, which may affect their purchase of, and access to, nutritious foods. Therefore, it is recommended, and supported by other recent research [35], that food literacy programs acknowledge the barriers faced by regional participants, and contextualise lesson content to their environmental and individual determinants in order to elicit positive behaviour outcomes, as already achieved by FSA. Overcoming these barriers are an important step towards achieving United Nations Goal 2: End hunger, achieve food security and improved nutrition and promote sustainable agriculture.

## Figures and Tables

**Figure 1 ijerph-18-08920-f001:**
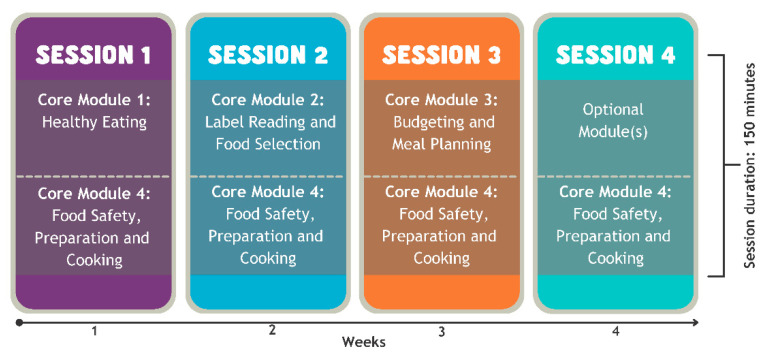
*Food Sensations* for Adults program structure.

**Table 1 ijerph-18-08920-t001:** Demographic characteristics of participants from metropolitan and regional areas.

Characteristic	Responses	FSA Respondents: Metropolitan (n = 1398)	FSA Respondents: Regional (n = 451)	*p*-Value
Sex		n = 1202	n = 451	<0.05
	Male	259 (21.5%)	62 (14.8%)	
	Female	943 (78.5%)	358 (85.2%)	
Age		n = 1201	n = 424	0.20
	18–25 y	165 (13.7%)	54 (12.7%)	
	26–35 y	298 (24.8%)	90 (21.2%)	
	36–45 y	284 (23.6%)	94 (22.2%)	
	46–55 y	149 (12.4%)	71 (16.7%)	
	56–65 y	147 (12.2%)	59 (13.9%)	
	66 y and over	158 (13.2%)	56 (13.2%)	
Household composition		n = 1193	n = 423	<0.05
	Couple with children	445 (37.3%)	131 (31.0%)	
	Single person	193 (16.2%)	71 (16.8%)	
	Partner	185 (15.5%)	99 (23.4%)	
	Single parent with child/children	118 (9.9%)	44 (10.4)	
	Other: that is family/extended family/shared accommodation	252 (21.1%)	78 (18.4%)	
Education level		n = 1189	n = 419	<0.001
	Certificate/Diploma/Trade	401 (33.7%)	134 (32.0%)	
	Finished high school	258 (21.7%)	122 (29.1%)	
	Bachelor degree or higher	333 (28.0%)	55 (13.1%)	
	Some secondary school or less	197 (16.1%)	108 (25.8%)	
Employment status		n = 1188	n = 418	0.10
	Unemployed/unable to work	375 (31.6%)	121 (28.9%)	
	House duties/maternity leave/retired	429 (36.1%)	134 (32.1%)	
	Part-time/casual	258 (21.7%)	109 (26.1%)	
	Full-time/self-employed	126 (10.6%)	54 (12.9%)	
Socioeconomic Index ^1^		n = 1146	n = 411	<0.001
	Low	435 (38.0%)	248 (60.3%)	
	Middle	314 (27.4%)	146 (35.5%)	
	High	397 (34.6)	17 (4.2%)	
Born in Australia ^2^		n = 1110	n = 397	<0.001
	Yes	588 (53.0%)	288 (72.5%)	
	No	522 (47.0%)	109 (27.5%)	
Identify as Aboriginal or Torres Strait Islander ^2^		n = 1103	n = 392	<0.001
	Yes	57 (5.2%)	52 (13.3%)	
	No	1046 (94.8%)	340 (86.7%)	

^1^ SEIFA, Socio-Economic Indexes for Areas, derived from postcode [31]. ^2^ Added in later version of questionnaire. Note: participants can have completed baseline only, post only or baseline and post.

**Table 2 ijerph-18-08920-t002:** Two-tailed t-tests comparing metropolitan vs. regional pre-program scores for the three food literacy behaviour domains and change in self-reported dietary intake of fruits and vegetables at baseline.

	Metropolitan Pre-Program (Mean)	Regional Pre-Program (Mean)	*p*-Value	95% CI of Difference-Lower	95% CI of Difference-Upper	% Difference
Food literacy behaviours (n = metropolitan, regional)						
PPlan and Manage (n = 1114, 389)	8.83	9.11	<0.05	−0.51	−0.05	3.17
PSelection (n = 1156, 410)	2.95	2.78	<0.01	0.04	0.30	−5.76
PPreparation (n = 1148, 405)	6.24	6.56	<0.001	−0.50	−0.13	5.13
Dietary intake behaviours (n = metropolitan, regional)						
PServes of fruit (n = 1072, 386)	1.58	1.55	0.56	−0.08	0.16	−1.90
PServes of vegetables (n = 1063, 386)	2.23	2.52	<0.0001	−0.43	−0.14	13.00

**Table 3 ijerph-18-08920-t003:** Self-reported fast food meal and sugar-sweetened drink intake frequency at baseline.

	Metropolitan Pre-Program	Regional Pre-Program	*p*-Value
Fast food meal frequency	n = 1069	n = 390	<0.05
Never	287 (26.8%)	130 (33.3%)	
Less than once a week	425 (39.8%)	155 (39.7%)	
Once or twice a week	297 (27.8%)	87 (22.3%)	
Three or more times a week	60 (5.6%)	18 (4.6%)	
Sugar-sweetened drink frequency	n = 1071	n = 389	<0.05
Never	509 (47.5%)	190 (48.8%)	
Less than once a week	263 (24.6%)	68 (17.5%)	
Once or twice a week	156 (14.6%)	65 (16.7%)	
Three or more times a week	82 (7.7%)	37 (9.5%)	
Five or more times a week	61 (5.7%)	29 (7.5%)	

**Table 4 ijerph-18-08920-t004:** Food literacy and dietary behaviours differences at baseline.

	Metropolitan Pre-Program	Regional Pre-Program	*p*-Value
Responsibility for meals	n = 1185	n = 418	0.126
All the time	682 (57.6%)	260 (62.2%)	
Shared	412 (34.8%)	136 (32.5%)	
No	91 (7.7%)	22 (5.3%)	
Responsibility for shopping	n = 1185	n = 414	0.020
All the time	640 (54.0%)	252 (60.9%)	
Shared	440 (37.1%)	139 (33.6%)	
No	105 (8.9%)	23 (5.6%)	
Cooking skills rating	n = 1188	n = 417	0.001
Can cook almost anything	282 (23.7%)	122 (29.3%)	
Can cook a wide variety of meals	496 (41.8%)	191 (45.8%)	
Can cook basic meat and 3 vegetables	296 (24.9%)	82 (19.7%)	
Can do basic heating food, use barbeque, boil egg	76 (6.6%)	10 (2.4%)	
Cannot cook or do not cook	35 (2.9%)	12 (2.9%)	
Healthy foods cost more	n = 1180	n = 411	0.010
Strongly disagree	91 (7.7%)	19 (4.6%)	
Disagree	330 (28.0%)	91 (22.1%)	
Not sure	239 (20.3%)	87 (21.2%)	
Agree	375 (31.8%)	161 (39.2%)	
Strongly agree	145 (12.3%)	53 (12.9%)	
Run out of money for food in the past month	(n = 1154)	(n = 406)	0.267
Never	705 (61.1%)	236 (58.1%)	
Sometimes	352 (30.5%	138 (33.9%)	
Most of the time	55 (4.7%)	23 (5.6%)	
Always	42 (3.6%)	9 (2.2%)	

**Table 5 ijerph-18-08920-t005:** Paired t-tests comparing metropolitan pre- and post-program scores for the three food literacy behaviour domains and change in self-reported dietary intake of fruits and vegetables.

	Metropolitan Pre-Program (Mean)	Metropolitan Post-Program (Mean)	*p*-Value	95% CI of Difference-Lower	95% CI of Difference-Upper	% Difference
Food literacy behaviours						
Plan and Manage (n = 719)	8.84	9.83	**<0.0001**	−1.11	−0.87	11.20
Selection (n = 780)	2.96	3.73	**<0.0001**	−0.86	−0.68	26.01
Preparation (n = 766)	6.22	7.10	**<0.0001**	−0.98	−0.77	14.15
Dietary intake behaviours						
Serves of fruit (n = 760)	1.59	1.86	**<0.0001**	−0.33	−0.20	16.98
Serves of vegetables (n = 756)	2.23	2.89	**<0.0001**	−0.75	−0.68	29.60

The bold shows the statistically significant results.

**Table 6 ijerph-18-08920-t006:** Paired t-tests comparing regional pre- and post-program scores for the three food literacy behaviour domains and change in self-reported dietary intake of fruits and vegetables.

	Regional Pre-Program (Mean)	Regional Post (Mean)	*p*-Value	95% CI of Difference-Lower	95% CI of Difference-Upper	% Difference
Food literacy behaviours						
Plan and Manage (n = 254)	9.29	9.79	**<0.001**	−0.68	−0.32	5.38
Selection (n = 270)	2.80	3.43	**<0.001**	−0.77	−0.48	22.50
Preparation (n = 256)	6.68	7.06	**<0.001**	−0.55	−0.20	5.69
Dietary intake behaviours						
Serves of fruit (n = 253)	1.56	1.71	**0.0171**	−0.26	−0.03	9.62
Serves of vegetables (n = 253)	2.60	2.87	**0.0002**	−0.41	−0.13	10.28

The bold shows the statistically significant results.

**Table 7 ijerph-18-08920-t007:** Two-tailed t-tests comparing metropolitan vs. regional post-program scores for the three food literacy behaviour domains and change in self-reported dietary intake of fruits and vegetables.

	Metropolitan Post-Program (Mean)	Regional Post (Mean)	*p*-Value	95% CI of Difference-Lower	95% CI of Difference-Upper	% Difference
Food literacy behaviours (n = metropolitan, regional)						
Plan and Manage (n = 952, 298)	9.76	9.71	0.6610	−0.18	0.28	−0.51
Selection (n = 997, 306)	2.67	3.43	**0.0017**	0.09	0.28	−6.54
Preparation (n = 987, 297)	7.08	7.00	0.3966	−0.11	0.28	−1.13
Dietary intake behaviours (n = metropolitan, regional)						
Serves of fruit (n = 985, 299)	1.84	1.71	0.0560	−0.00	2.25	−7.07
Serves of vegetables (n = 981, 299)	2.84	2.88	0.6649	−0.20	0.13	1.41

The bold shows the statistically significant results.

## Data Availability

The data presented in this study are available on request from the Western Australia Department of Health. The data are not publicly available due to contractual arrangements.

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
