# Peer review of "Effectiveness of Foodbank Western Australia’s Food Sensations® for Adults Food Literacy Program in Regional Australia"

_ijerph, 2021, doi:10.3390/ijerph18178920_

Round 1

Reviewer 2 Report

The aim of this research was to determine if there are differences in the effectiveness of Food Sensations® for Adults (FSA)  in regional and metropolitan participants in Australia. The study showed that there were no differences in food literacy behaviors over a four-week period in both regional and metropolitan participants. 

While the work has been carried out in a formally correct manner there are shortcomings that prevent its publication in its current form.

  1. There are many similarities between this paper and others published by the same research group (please also look at the attached file). 

    https://onlinelibrary.wiley.com/doi/10.1111/1747-0080.12627

    https://ro.ecu.edu.au/cgi/viewcontent.cgi?amp=&article=10976&context=ecuworkspost2013

  2. The number of self-citations is exaggerated
  3.  Food Sensations® sounds a lot like a regular nutrition education program. What are the differences? What are the characteristics of the nutrition plan that is conveyed to the subjects? 
  4. it would be necessary to conduct a study in which a guideline-based nutrition education plan is compared to Food Sensations®. 

Reviewer 3 Report

Language and technical care:

The level of care shown in respect of language use and editing in this paper is commendable – the paper reads well, and the authors should be congratulated for such an excellent piece of work. There are just a few minor suggestions:

  • Page 2, line 73 – the reviewer is uncertain about the sentence that reads '…who would like to increase their food literacy skills.' Is food literacy something that the man in the street is aware of and would they ever feel the need to want to improve their food literacy? The reviewer is also questioning if one should refer to 'food literacy skills', or rather ‘level of food literacy'. Is food literacy a skill?
  • Page 3 – Figure 1 at the top of the page should be moved slightly to the right to line up with the text underneath.
  • Page 3 & 4 – lines 123 to 133 – this paragraph is not justified on the right-hand side of the page like the rest of the document;
  • Page 10 – line 282 – there is a closing bracket missing after the p value – p=0.12170).;
  • Page 11 – line 359 – there is an additional space before the square reference brackets [43] which needs removing;
  • Page 12 – line 375 and 377 – in the discussion of limitations the authors state that there are several limitations, starting with ‘Firstly’, but then later ‘Secondly’, and then another ‘Secondly’ in line 377.

The manuscript is very well referenced using relevant, up-to-date references.

Literature Review:

This reviewer identified no shortcomings in the literature review. On page 10, lines 307 and 308 a slightly better explanation of the expensive nature of healthy food in rural areas would greatly enhance the understanding of this concept.

Methodology and materials:

This reviewer believes that the methodology was clearly explained and followed according to acceptable research procedures. There was no uncertainty of the processes followed in terms of identifying the sample, collecting data nor the analysis and interpretation of data.

Results and Discussion:

The reviewer believes that the results are of high quality and value, particularly in context of the growing nature of eating behaviour and consequent food literacy levels.

Conclusion:

This reviewer believes that the conclusion is well written. In a time where many people on the planet suffer from under- or malnutrition, all work to end this preventable situation, or support efforts to eradicate poverty and hunger is worthy research. The authors may have considered contextualising their work in the larger global context, such as the United Nation’s SDG Goal 2 which is to “End hunger, achieve food security and improved nutrition and promote sustainable agriculture”.

Round 2

Reviewer 2 Report

The authors have made some improvements to the paper. Despite this, the issues reported in my previous review are still present. In particular there are no major differences between Food Sensations® and a normal nutrition education program. 

Author Response

Author’s responses

The authors are agreeing with Reviewer 2 that Food Sensations is a nutrition education program.  We have indicated that it is advertised to the public in Western Australia as a nutrition education and cooking program (Line 75) with the aim of improving food literacy.  Nutrition Education programs can take many different forms.  The authors agree that Food Sensations is a nutrition education program as indicated in Round 1 reviews, but that the program’s focus is to developing food literacy knowledge and skills.

The special edition of the journal that this paper is being considered for is titled Food Literacy and Public Health.  Therefore we have indicated that Food Sensations is a nutrition education program focusing on food literacy.

In addition, we have now changed the abstract line 11 to indicate Food Sensations is a four-week nutrition education program focused on food literacy.

In addition, we have now changed food literacy interventions on line 63 and 65 to ‘nutrition education programs focusing on food literacy’ and ‘programs’.

We have replaced the term ‘intervention’ with ‘program’ or ‘nutrition education program’ where required. 

The objective of the paper is to compare the impact on participants in regional Australia (Line 128-132).  It is not the aim of the paper to establish differences between other nutrition education programs.  We have compared Food Sensations to other similar nutrition education programs in the discussion where appropriate, but acknowledge this was challenging due to the limited number of published programs (Line 353-354).
